# Thoracic Surgery in the COVID-19 Pandemic: A Novel Approach to Reach Guideline Consensus

**DOI:** 10.3390/jcm10132769

**Published:** 2021-06-24

**Authors:** Tomasz Dziodzio, Sebastian Knitter, Helen Hairun Wu, Paul Viktor Ritschl, Karl-Herbert Hillebrandt, Maximilian Jara, Andrzej Juraszek, Robert Öllinger, Johann Pratschke, Jens Rückert, Jens Neudecker

**Affiliations:** 1Department of Surgery, Campus Charité Mitte and Campus Virchow-Klinikum, Charité-Universitätsmedizin Berlin, 13353 Berlin, Germany; sebastian.knitter@charite.de (S.K.); helen-hairun.wu@charite.de (H.H.W.); paul.ritschl@charite.de (P.V.R.); karl-herbert.hillebrandt@charite.de (K.-H.H.); maximilian.jara@charite.de (M.J.); robert.oellinger@charite.de (R.Ö.); johann.pratschke@charite.de (J.P.); jens-c.rueckert@charite.de (J.R.); jens.neudecker@charite.de (J.N.); 2BIH Charité Clinician Scientist Program, Berlin Institute of Health (BIH), 10178 Berlin, Germany; 3Department of Cardiac Surgery and Transplantation, The Cardinal Stefan Wyszyński National Institute of Cardiology, 04-628 Warsaw, Poland; anderso@o2.pl

**Keywords:** SARS-CoV-2, COVID-19, thoracic surgery, recommendations, guidelines

## Abstract

The COVID-19 pandemic challenges international and national healthcare systems. In the field of thoracic surgery, procedures may be deferred due to mandatory constraints of the access to diagnostics, staff and follow-up facilities. There is a lack of prospective data on the management of benign and malignant thoracic conditions in the pandemic. Therefore, we derived recommendations from 14 thoracic societies to address key questions on the topic of COVID-19 in the field of thoracic surgery. Respective recommendations were extracted and the degree of consensus among different organizations was calculated. A high degree of consensus was found to temporarily suspend non-critical elective procedures or procedures for benign conditions and to prioritize patients with symptomatic or advanced cancer. Prior to hospitalization, patients should be screened for respiratory symptoms indicating possible COVID-19 infection and most societies recommended to screen all patients for COVID-19 prior to admission. There was a weak consensus on the usage of serology tests and CT scans for COVID-19 diagnostics. Nearly all societies suggested to postpone elective procedures in patients with suspected or confirmed COVID-19 and recommended constant reevaluation of these patients. Additionally, we summarized recommendations focusing on precautions in the theater and the management of chest drains. This study provides a novel approach to informed guidance for thoracic surgeons during the COVID-19 pandemic in the absence of scientific evidence-based data.

## 1. Introduction

For more than a year, the severe acute respiratory syndrome coronavirus 2 (SARS-CoV-2) pandemic has challenged healthcare providers worldwide [1]. Strict containment rules have been introduced in many countries, including social distancing, stay-at-home orders and mandatory face masks in public life. Simultaneously, hospitals are faced with unprecedented issues regarding the adequate allocation of resources between the acute care of COVID-19 patients and sustaining the management of non-COVID-19 patients. Shortages in staff, diagnostics and hospital capacity have led to the abrupt reduction or even halting of all surgical procedures. In the case of thoracic surgery, the impact of the novel SARS-CoV-2 is even more delicate: First, patients who undergo thoracic surgery may be prone to poor outcomes in association with COVID-19. In addition, due to the nature of thoracic surgery focusing on lung diseases, exposure of hospital staff to SARS-CoV-2 may be increased. Therefore, clear guidelines and recommendations are essential for healthcare providers. The aim of this study was to provide a summary of available expert guidelines for managing thoracic surgery programs during the ongoing COVID-19 pandemic until evidence-based guidelines become available.

## 2. Materials and Methods

### 2.1. Study Design 

Beginning in March 2020, recommendations of national and international thoracic surgery societies on COVID-19 were identified by an online search from official websites. Only recommendations published or referenced by thoracic surgery societies were considered. Societies with recommendations that were not publicly available were contacted by email. The acquired recommendations were pooled and examined by two independent researchers (TD and SK). Key statements were extracted and assigned to defined thematic areas. Statements were categorized and the degree of consensus was evaluated. Categories were defined as (a) does support (+), (b) does not support (-), (c) leaves the answer open/case-by-case decision (±) or (d) does not comment (n/a) [2]. In case of diverging classification, a third investigator (JN) was consulted. 

In a second step, extracted key statements were sent to all previously identified national and international thoracic surgery societies in order to achieve a comprehensive consensus of the recommendations. The key statements and responses of the societies were summarized. A score was calculated by subtracting the number of negative recommendations from the count of positive recommendations. Zero points were counted if the answer was left open or no statement was available. The society recommendation consensus (SRC) was classified as “strong recommendation” (SRC = A, score > 10, societal approval of >70%), “medium recommendation” (SRC = B, score 4 to 9, societal approval of 25–69%) or “low recommendation” (SRC = C, score 1 to 3, societal approval of <25%). Subsequently, the statements were discussed on the basis of existing literature. No study approval was necessary, as all information was publicly available, or if not online, provided by the societies.

### 2.2. Statistics

No statistical tests were used to reach consensus. The study did not compare strategies of different countries, nor was it used to draw causal conclusions. 

## 3. Results

Nine society recommendation bulletins on thoracic surgery in the COVID-19 pandemic were identified as of 1st March 2021 (out of 22 checked societies; countries with society names in alphabetical order in brackets): Australia/New Zealand (ANZSCTS/RACS), Brazil (SBCT), China (CSTCS), France (SFCTVCS), Germany (DGT), Italy (SICT), Spain (SECT), United Kingdom (SCTC/RCS), USA (AATS/ACS). The Austrian Society of Thoracic Surgeons (OGTC) had no published guidelines and referred to the guidelines of the DGT. The European Association of Cardio-Thoracic Surgeons (EACTS) and the European Society of Thoracic Surgeons (ESTS) referred to national guidelines of each European society. The World Health Organization (WHO) had no published recommendations for thoracic surgeons or patients on its website by 1st March 2021. 

Conforming with the information given on the individual websites (Table 1), 27 recommendation statements were extracted and summarized under six subheadings (Figure 1):General StatementsStaffPrecautions in the theaterDiagnosticsTreatmentChest drains

Five societies replied to the statement proposal request (response rate 23%). Their assessments were taken into account in the consensus-building process.

### 3.1. Statements

#### 3.1.1. General Statements

The guidelines and recommendations should be adapted according to the local prevalence of COVID-19 and the hospital’s resources (SRC A).

The incidence of COVID-19 differs substantially around the world with various epicenters [1]. Among all guidelines, a strong consensus was found for the adaptation of recommendations according to the local prevalence of COVID-19. Maintaining and providing thoracic surgeries in this context requires good preplanning and vigilance against infection control measures at all levels. The objective should be a compromise between the surgical treatment of thoracic pathologies and preservation of hospital resources for COVID-19 patients. 

2.All patients for whom the delay of surgical procedures is necessary should be tracked and their treatment should be prioritized. The usage of alternative treatment options should be considered and documented (SRC A).3.If standard care (e.g., resection) is not available, an individual treatment plan should be made for each patient by a multidisciplinary team (SRC A).

With increasing numbers of COVID-19 cases, elective surgical procedures may be halted by local, regional and state-based authorities. Thoracic surgery includes a variety and elective cases and emergencies. Therefore, a deliberate prioritization of surgical cases should be considered and stratified in emergency, urgent and elective procedures [3,4]. The latter can be further divided into high, medium and low priority [5]. This classification may help to prioritize patients depending on the hospital´s resources and current pandemic trajectory. For cancer patients, postponing cancer resection may considerably influence short-term and long-term survival. Delays for up to four weeks may not be associated with a significant impact on patient survival [6]. A strong consensus was found for establishing a list of patients who had their surgery delayed with constant reevaluation. 

4.The attendance of patients in the hospital should be limited. Family visits should be reduced to one or no visitors (SRC A).

Cancer patients located in an epicenter of a viral epidemic harbor a higher risk of SARS-CoV-2 infection compared with the community [7]. Thus, patients should only be present at hospitals if truly necessary. Visitation should be suspended or heavily restricted and only allowed in case of life-threatening conditions.

5.On arrival at the clinic, all patients should wear surgical masks (SRC A).6.Staff should wear surgical masks at any patient contact (SRC A).

Strong agreement was detected with respect to surgical masks for hospital patients in general and that staff members should wear surgical masks when in contact with patients. Clear evidence on the effectiveness of surgical masks is still lacking and contradicting reports in the literature exist [8,9,10]. Recent studies support using masks by suggesting that they could save lives in different ways: By cutting down the chances of transmitting and contracting COVID-19 [11]. Furthermore, some studies hint that wearing a mask might also reduce the severity of infection [12].

#### 3.1.2. Staff

7.Staffing should be kept to a minimum. Virtual appointments/conferences and consultations should be preferred (SRC A).

Decreasing interactions between staff members and workforce distribution can reduce the risk of transmission within the division [9,10]. Working from home should be recommended for all non-clinical staff members if possible [13]. Telephone and/or video conferences should be encouraged. 

8.Staff members should not be screened for SARS-CoV-2 (SRC C).

Two societies recommended to screen all hospital staff for SARS-CoV-2 and four societies disagreed on the topic. Some societies recommended nasopharyngeal swab tests in staff presenting with typical symptoms of COVID-19. In such a condition, self-isolation at home needs to be started immediately. In case of a negative test result, staff should resume work [14]. In recent literature, some authors highlight the benefits of routine testing of healthcare workers in reducing the asymptomatic spread of SARS-CoV-2 [15] and describe universal testing as feasible [16], while others fail to see the indication if adequate protective gear is used [17].

#### 3.1.3. Screening of Patients

9.All patients should be evaluated for respiratory symptoms before hospitalization (SRC A).10.All patients should be screened for SARS-CoV-2 by nasopharyngeal swabs (SRC C).

Most societies agreed that patients’ previous medical history should be screened for typical COVID-19 symptoms and physical examination should be conducted [18,19]. However, only a few societies that provided COVID-19 guidelines gave the recommendation to establish routine SARS-CoV-2 testing for all patients by using nasopharyngeal swabs. Some authors even reported high false-negative rates of RT-PCR [20,21,22]. More recent data support routine testing of patients and staff as prices, availability and detection quality improve [23].

If routine testing of patients is performed, the following procedure is recommended: (1) Testing at admission in the hospital, (2) isolation with appropriate protection until obtaining test results, (3) in case of emergency surgery, behavioral instructions for COVID-19-positive patients should be applied until confirmation of a negative result.

11.Serology tests are generally recommended (SRC C).

As of now, several antibody tests have been approved by the United States Food and Drug Administration and the European Union [24]. The adoption of antibody-directed tests has not been recommended by any thoracic surgery society to date and a low consensus was found on the use of serology tests. While the literature agrees that serological tests may provide an essential tool in managing the pandemic [25,26], recent data from a meta-analysis emphasize the need for more evaluation of the diagnostic accuracy and the optimal timing of serological tests [27,28,29].

12.Preoperative CT scans should be conducted for all cancer surgery patients that require critical care (IMC/ICU) postoperatively (SRC C).

It has been proposed that patients planned for intensive care monitoring after elective thoracic surgery receive a CT scan preoperatively [30]. The aim of this procedure is to identify patients with pathological changes typical of COVID-19 before results of RT-PCR are available, or in the absence of SARS-CoV-2 test kits. However, the sensitivity of CT scans in diagnosing COVID-19 remains debatable. While some studies report sensitivity rates of up to 98% [31] and 97% [32], others give false-negative rates of 20% [33], or criticize methodological flaws [34]. Bernheim et al. reported that 56% of patients with COVID-19 had a normal CT in the first two days [35]. A Cochrane review found a sensitivity of chest CT of 90%, and a false-positive rate of 38% [36]. Thus, only a low level of recommendation can be given for this statement at this point.

#### 3.1.4. Precautions in the Theater

13.A designated theater and scrub room should be used for suspected or proven COVID-19 patients. A preoperative COVID-19 checklist should be used for suspected and confirmed COVID-19 patients (SRC B).

It has been proposed to establish a theater designated for proven or suspected COVID-19 cases in order to create a physical separation. Detailed guidelines and a preoperative checklist have been provided by the SCTS [37]. 

14.During procedures with suspected or proven COVID-19 patients, no changes in staff should be made. Reduce personnel to a minimum. Non-essential personnel should be absent (e.g., medical students and nurses-to-be) (SRC B).

A medium consensus was reached for this statement. In order to reduce the possibility of infection and to minimize transmission of aerosolized SARS-CoV-2, most societies recommend that staffing levels should remain constant through the surgery and these procedures should not be used for training purposes.

15.Appropriate PPE (≥PPE2) should be used for all patients and in case of COVID-19-positive patients: PPE2/3 and goggles (SRC A).

A strong recommendation was given to wear at least personal protective equipment (PPE) category 2 for all thoracic procedures. In case of COVID-19-positive patients, PPE2/3 and goggles should be used. The numbers indicate three different categories as defined by the European Union (Council Directive 89/686/EEC): “PPE1” refers to basic protective equipment with a simple design, whereas complex designs are included under “PPE3” (e.g., respiratory equipment). “PPE2” refers to all PPE not falling into categories 1 or 3. Due to the significant environmental contamination of SARS-CoV-2, appropriate protection is an absolute must [38]. 

16.The use of laminar airflow is recommended in the theater (SRC C).

Low consensus was reached on this statement. A general recommendation to use ventilation techniques to reduce the risk of transmission of SARS-CoV-2 in operating or procedural rooms could not be given, mostly because of a lack of evidence [39]. Still, several authors recommend negative pressure environments as a preventive measure for protection against SARS-CoV-2 [40,41,42,43].

#### 3.1.5. Diagnostics

17.The routine use of low-dose CT scans instead of chest X-ray (CXR) is not recommended (SRC C).

There is only a low consensus on the use of CT scans instead of CXRs as a routine diagnostic procedure. No evidence was found that in the COVID-19 pandemic should a CT scan be performed if the needed information can be provided by CXR [44]. 

18.Bronchoscopy should only be performed in patients who have no symptoms, contact or imaging suggestive of COVID-19 infection and postponed in patients with suspected or confirmed COVID-19 infections (SRC A).

There is little evidence on the role of bronchoscopy during the COVID-19 pandemic [45]. Most societies recommend postponing bronchoscopies if possible, or an interval of 28 days from the onset of infection. However, if bronchoscopy is unavoidable, it should be performed under increased precautions in both symptomatic and asymptomatic patients. 

19.Avoid high-flow nasal oxygen or aerosol-generating procedures (SRC A).

There is a strong agreement on the avoidance of high-flow nasal oxygen (HFNO) and aerosol-generating procedures during diagnostic procedures due to fear of aerosol dispersion. However, recent evidence on aerosol generation associated with HFNO fails to establish a clear link between the use of HFNO and increased SARS-CoV-2 infection rates of healthcare workers [46,47]. In addition, HFNO has been proven to reduce intubation rates and overall mortality in patients with COVID-19 [48,49]. Therefore, proven benefits and potentially unknown risks associated with HFNO must be carefully balanced. Aerosol dispersion may be reduced by using a surgical mask over the face of the patient on HFNO [50].

#### 3.1.6. Treatment

20.Triage and surgical indications should be adapted according to the local prevalence of COVID-19 and the hospital’s resources (SRC A).

All societies have strongly agreed to a three-phase triage approach. Depending on the national administrative structure, phase assessment should be conducted following a protocol with respect to the number of COVID-19 patients within the hospital (A), the availability of hospital resources (B) and the dynamic ratio between A and B. Since the course of the COVID-19 pandemic remains unclear, all societies point out that triage recommendations should not be considered as rigid guidelines. They should be applied individually to each patient, as the situation may change rapidly. All depicted societies embedded the proposed triage guidelines either directly or by reference in their recommendations [51] (Figure 2). However, a huge number of elective surgical procedures already have been canceled or postponed, risking untreated or inappropriately treated lung cancer [52]. Therefore, constant reevaluation of patient selection guidelines and differential diagnosis for benign entities are necessary to accurately select patients undergoing lung cancer surgery and ensure a rapid diagnostic and therapeutic process.

21.Patients with symptomatic or more advanced cancers should be prioritized for surgery (SRC A).

In particular, patients with a predicted decrease in survival are considered at risk and should be prioritized. This is supported by evidence from national cancer registries [53,54]. Additionally, a recent study by Chang et al. found that pulmonary resection for lung cancer can be safely performed in selected patients, even when performed in a hospital with a large COVID-19 census [55]. 

22.Surgery for non-critical elective/benign conditions should be postponed (with constant reevaluation) (SRC A).

The decision on the treatment of elective/benign conditions should be made according to the national triage classification with close monitoring and reevaluation. However, it should be noted that these patients are at higher risk of acquiring infections, including COVID-19, due to their underlying disease or previous chemotherapies [56,57]. These patients are particularly at risk for prolonged ICU stay, necessity for mechanical ventilation and death [58].

23.In patients with proven COVID-19 infection, only essential and life-saving surgeries should be performed after a multidisciplinary decision (SRC B).

Hospital-associated transmission is a relevant mechanism of COVID-19 infection of healthcare professionals (29%) and hospitalized patients (12.3%) [59]. All non-essential surgical procedures on COVID-19 positive patients should be postponed until confirmed infection convalescence. 

#### 3.1.7. Chest Drains

24.The use of a closed system connected to a bag instead of a water seal system is recommended in pleural effusions (SRC C).25.In pneumothorax with indication of thoracic drainage, it is recommended to connect to a conventional water seal system (SRC B).26.There is no benefit of digital drain systems (SRC B).

Experts reached a low to medium consensus regarding the handling of chest tubes in the COVID-19 pandemic. The utilization of chest drains may be associated with a higher risk of aerosolization. There is still uncertainty about which chest drain system is best to avoid or decreases aerosol generation. Proper preparation, PPE, modified techniques and drainage maintenance are critical to minimizing exposure of healthcare personnel [60,61]. Therefore, it is recommended to use closed systems (digital), anti-viral filters or connect chest drains with water seals to wall suction (even in cases where suction is not indicated and set at a very low, controlled level, e.g., 2–5 cmH2O) to reduce aerosol building up. Some societies additionally recommend performing thoracic tube placement by a dedicated thoracostomy team [62] or the use of high-efficiency particulate air (HEPA) filters [63].

27.Avoid early removal of chest drains placed in patients with COVID-19 infection and pneumothorax. Drains should be closed at least 24 hours before radiological confirmation and removal (SRC C).

To prevent repetitive chest tube placements, three societies proposed performing a CXR after a 24 h interval of clamping before removal of chest tubes.

## 4. Discussion

Healthcare professionals are currently facing unprecedented challenges in handling the COVID-19 pandemic-induced healthcare crisis. Scientific evidence is scarce and thus strategies are mostly based on expert opinion rather than evidence. In the field of thoracic surgery, many questions remain unclear. Thoracic surgeons must deal directly with the clinical picture of the COVID-19 disease, and it is becoming more and more apparent, in particular the consequences for the rest of the patient population.

The core findings were the following: Action plans can be based on these proposed recommendations and be adapted according to the local prevalence of COVID-19 and the hospital’s resources (SRC A). If standard care is not available, an individual treatment plan should be made for each patient with a multidisciplinary team (SRC A). All patients for whom the delay of surgical procedures is necessary should be tracked and prioritized later on with constant reevaluations (SRC A). When surgical therapies are delayed and access to medicine is limited, the implementation of prehabilitation programs (PREHAB) and enhanced recovery after surgery (ERAS) protocols for affected patients may help in overcoming the drawbacks of the pandemic [64]. Nutritional interventions and physical activities can help to improve the physical condition of patients until elective surgical therapies become available again. Patients with malignant diseases are particularly affected, as their survival depends heavily on early initiation of state-of-the-art treatment. All societies agreed that patients with symptomatic or more advanced cancer should be prioritized for surgery, especially if predicted patient survival is significantly impaired (SRC A). 

With respect to staff and patient precautions, all societies agreed on reducing contact between patients and staff by implementation of video appointments (SRC A). Face masks should be worn as long as the pandemic continues by patients and hospital stuff at any contact (SRC A). Low consensus was found on screening all personnel and patients for SARS-CoV-2 (SRC C). It is worth mentioning that many societies have not updated their recommendations for some time and that screening of personnel and inpatients has now become standard practice due to the broad availability of tests and, therefore, this recommendation seems outdated.

Additionally, it is recommended to have a separate operating room track for COVID-19 patients (SRC B) and procedures performed on patients with suspected or confirmed COVID-19 should be kept as short as possible. Albeit only a moderate consensus was found on the usage of closed chest tube systems or anti-viral filters (SRC B/C), more recent data and publications support the usage of closed systems to reduce aerosolization to a minimum and to use a specialized thoracostomy team [60,62].

Generating a “conventional” consensus usually requires existing data and is a very time-consuming process requiring repeated reevaluations and discussions. Since evidence and time are still lacking, we based our novel consensus-building approach on expert guidelines and focused on recommendations with definitive statements on the depicted topics. Our study can serve as a basis for future decision-making until evidence-based results become available. However, since we bypassed the elaborate consensus-building process, some relevant limitations need to be addressed. The first major limitation of this work is the distortion caused by interpretation and translation errors. Occasional re-interpretation of societal guidelines was necessary to confirm or dismiss the proposed key statements. Due to the lack of scientific evidence, the abovementioned consensus statements are based mostly on expert opinions as the retrieved information was not intended to serve as a basis for consensus recommendation. This limitation is aggravated by the low response rate of 23% after contacting thoracic surgery societies. By answering directly to our proposed statements, interpretation errors could have been avoided and more clear deductions would have been possible. Additionally, we primarily focused on definitive statements and some less clear statements were listed rather than counted in the recommendation-building process. Therefore, some of our statements may underestimate and other overestimate certain topics. Decision making in the pandemic is not always straightforward and there is rarely one right way. Each country, each region and each hospital has different pandemic situations and also different resources to deal with it. There is no “one size fits all” solution and some recommendations may apply well for some readers other may not. Another weakness of this work is its moderate timeliness. The pandemic situation changes weekly, and some recommendations may already be outdated at the time of publication or new aspects may not be taken into account (e.g., screening of patients and staff). In addition, several societal recommendations have not been updated for months or adapted to recent scientific findings and the current pandemic situation. Consequently, all recommendations, even if supported by a high SRC, have to be classified as the lowest (scientific) evidence level and should be interpreted as such. Finally, this review is not a complete overview of all recommendations on the topic as similar societal recommendations from other disciplines (especially anesthesiology/internal medicine) were not analyzed.

## 5. Conclusions

In the quest for a rapid transition back to the field of evidence-based medicine, accurate documentation and publication of prospectively collected clinical data on COVID-19 in thoracic surgery patients are necessary. We suggest that these studies also should scrutinize the evidence of the abovementioned emphasized statements. In the meantime, our summary of expert recommendations in the field of thoracic surgery may help with decision making during this difficult time.

## Figures and Tables

**Figure 1 jcm-10-02769-f001:**
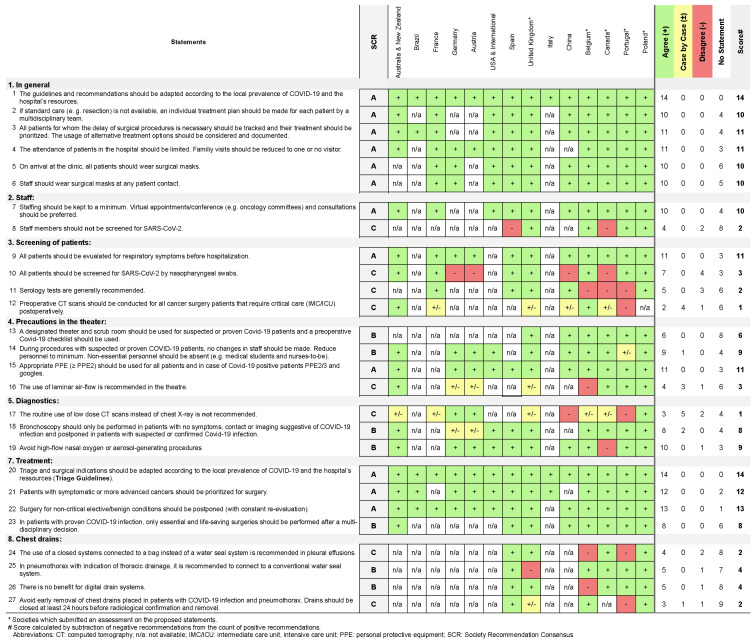
Proposed statements and SRC classification.

**Figure 2 jcm-10-02769-f002:**
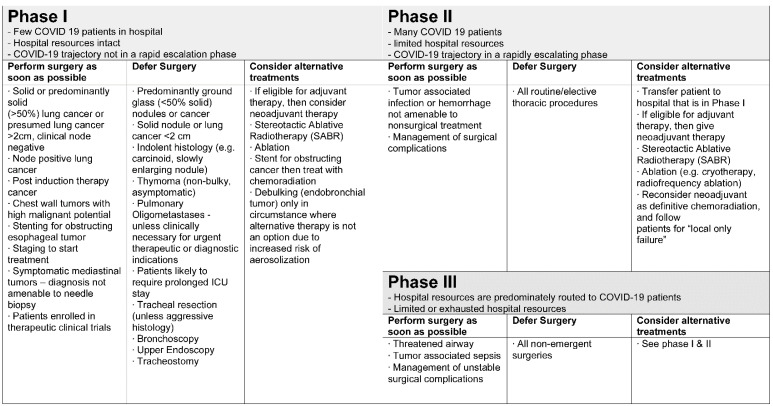
Triage compass based on the recommendations of the Thoracic Surgery Outcomes Research Network [51].

**Table 1 jcm-10-02769-t001:** Thoracic societies, access and recommendation date.

	Country	SOCIETY	Link	Date *	Recommendation Available Online
1	Australia and New Zealand	ANZSCTS	https://anzscts.org/COVID-19-resources/	04/07/2020	yes
2	Austria	OGTC	https://www.ogtc.at/	04/06/2020	no
3	Belgium	BACTS	https://www.bacts.org/	05/19/2020 †	no
4	Brazil	SBCT	https://www.sbct.org.br/recomendacoes-da-sociedade-brasileira-de-cirurgia-toracica-sbct-para-realizacao-de-traqueostomias-e-manejo-da-via-aerea-em-casos-suspeitos-ou-confirmados-de-infeccao-pelo-novo-coronavirus-c/	03/23/2020	yes
5	Canada	CATS	https://www.canadianthoracicsurgeons.ca/	05/19/2020†	no
6	China	CSTCVS	http://www.cstcvs.net/comsite/news/show/cn/3208.html http://www.cstcvs.net/comsite/news/show/cn/3208.html	02/26/2020	yes
7	Europe	EACTS ESTS	https://www.eacts.org/update-coronavirus-COVID-19/ http://www.ests.org/	04/27/2020	no
8	France	SFCTVCS	https://www.sfctcv.org/covid19-recommandations-de-la-sfctcv/	04/20/2020	yes
9	Germany	DGT	https://dgt-online.de/empfehlungen-des-dgt-vorstands-zur-corvid-19-pandemie/	04/06/2020	yes
10	International	STS	https://www.sts.org/COVID-19	07/03/2020	yes
11	Poland	PTKT	https://ptkt.pl/	05/18/2020 †	no
12	Portugal	SPCCTV	https://www.spcctv.pt/	05/18/2020 †	no
13	Spain	SECT	https://www.sect2020.pacifico-meetings.com/index.php/informacion-COVID-19	04/29/2020	yes
14	UK	SCTS RCS	https://scts.org/COVID-19/ https://www.rcseng.ac.uk/coronavirus/	04/22/2020 05/19/2020 †	yes
15	USA	AATS	https://www.aats.org/ https://www.jtcvs.org/COVID-19	04/07/2020	yes

* Date of last recommendation update; authors checked recommendation updates until March 1st 2021; † date of response to the proposed statements.

## Data Availability

No data reported.

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
