# Peer review of "Thoracic Surgery in the COVID-19 Pandemic: A Novel Approach to Reach Guideline Consensus"

_jcm, 2021, doi:10.3390/jcm10132769_

Round 1

Reviewer 1 Report

It's an interesting work. My consideration are as follow:

1) Please specify the period of study;

2) What is the COVID and COVID free path recommented after the study of different guidelines?;

3) I don't agree with your considerations about chest drain.  I think that the thoracostomy tube placement must be carried out by an experienced doctor or surgeon, assisted only by a staff member as recommended by the AAST. The use of closed digital chest drain system is necessary because it reduces the aerosolization of virus and the exposure of the team. An alternative method (HEPA filter) has been proposed by Carvalho et al. Please read the following articles:

  1. Pieracci FM, Burlew CC, Spain D, Livingston DH, Bulger EM, Davis KA, Michetti C. Tube thoracostomy during the COVID-19 pandemic : guidance and recommendations from the AAST Acute Care Surgery and Critical Care Committees. Trauma Surg Acute Care Open 2020; 5: e000498. doi: 10.1136/tsaco-2020-000498
  2. Carvalho EA, Oliveira MVB. Safety model for chest drainage in pandemic by COVID-19. Rev Col Bras Cir 2020; 47: e20202568. doi: 10.1590/0100-6991e-2020256

3) There are the recommendations about thoracic surgery?

Author Response

Dear Editorial Board and Reviewers,

Thank you very much for the valuable review and the possibility to provide a revised version of our manuscript "Thoracic surgery in the COVID-19 pandemic: A novel approach to reach guideline consensus" (Manuscript ID: JCM-1176808-R1) to be considered for publication in your esteemed Journal of Clinical Medicine as an original article.

First, we would like to thank the reviewers for the valuable comments, which have been very fruitful for the refinement of our manuscript. The entire manuscript was thoroughly revised according to the referees’ comments. A detailed point-by-point response is attached below. We assume that this revision has substantially improved the quality of the manuscript and hope that the reviewers and editorial board will agree.

All authors are in agreement with the content of the revised version. The content has not been published or is under consideration for publication elsewhere.

Thanking you in advance for your endeavors.

Yours sincerely,

Tomasz Dziodzio, MD

Point-by-point response to the comments:

Response to the summary of assessments by Reviewers:

Reviewer 1:

It's an interesting work. My consideration are as follow:

  • Please specify the period of study;

Response: We thank the reviewer for this comment. The online search for guidelines and recommendations began in March 2020 and was regularly repeated until an internally scheduled deadline of March 1st, 2021. We adapted the manuscript accordingly.

  • What is the COVID and COVID free path recommended after the study of different guidelines?

Response: We thank the reviewer for this helpful remark. Deducted from our research process, we recommend to separate “COVID-“ and “COVID-free” pathways according to the SARS-CoV-2 status in each respectable patient: For proven or suspected SARS-CoV-2 positive patients, preventive and precautionary measures should be performed as outlined in our statements (see Figure 1). In case of a recent negative (PCR) test, easing to some extent may be possible, e. g. regarding the use of specialized operating theatres.

3) I don't agree with your considerations about chest drain.  I think that the thoracostomy tube placement must be carried out by an experienced doctor or surgeon, assisted only by a staff member as recommended by the AAST. The use of closed digital chest drain system is necessary because it reduces the aerosolization of virus and the exposure of the team. An alternative method (HEPA filter) has been proposed by Carvalho et al. Please read the following articles:

Pieracci FM, Burlew CC, Spain D, Livingston DH, Bulger EM, Davis KA, Michetti C. Tube thoracostomy during the COVID-19 pandemic: guidance and recommendations from the AAST Acute Care Surgery and Critical Care Committees. Trauma Surg Acute Care Open 2020; 5: e000498. doi: 10.1136/tsaco-2020-000498

Carvalho EA, Oliveira MVB. Safety model for chest drainage in pandemic by COVID-19. Rev Col Bras Cir 2020; 47: e20202568. doi: 10.1590/0100-6991e-2020256

Response: We thank the reviewer for this comment. At the time of writing our manuscript, guidelines by societies from other specialties were not within the scope of our research process. However, we agree with your remarks and the statements in your references. We edited the section in our manuscript accordingly.

4) There are the recommendations about thoracic surgery?

Response: Thank you for the remark. At the time of drafting the manuscript, several thoracic surgery societies had published guidelines on the topic of COVID-19 (refer to Table 1). These recommendations included logistic considerations (specialized operating theatre, reduced staff presence), preventive measures (appropriate PPE), or diagnostic approaches. Of note, a triage compass classifying surgical indications according to the local prevalence of COVID-19 was also published (see Figure 2). To our knowledge, detailed statements or recommendations regarding individual procedures do not exist in the literature.

Reviewer 2 Report

Thankyou for the opportunity to review. I read with interest this manuscript attempting to distil some consensus from a broad range of specialist society guidelines concerning the practice of Thoracic surgery during the COVID-19 pandemic. The sheer quantity and breadth of guidelines and recommendations that have been rapidly generated in response to the pandemic has been vast, and I am therefore convinced that there is a real place for the type of summary publication presented, particularly where the methodology is robust.

Major comments

The manuscript brings together the recommendations of many international thoracic surgical societies, but regrettably has failed to acknowledge the existence of a similar number of anaesthetic / perioperative publications, the subject mater of which overlap significantly.

I strongly support the attempt to use robust methodology to distil consensus from a number of similar, but at times divergent sources. The methodology however is inadequately described. Further, the results of this process are inadequately described, and the reader is left uncertain in places as to how any particular recommendation has been reached. The following is a prime (but not the only) example:

3.1.2 Staff

  1. Staff members should not be screened for SARS-CoV-2.(SRC C) Two societies recommended to screen all hospital staff for SARS-CoV-2 and four so[1]cities disagreed on the topic. Some societies recommended nasopharyngeal swab tests in staff presenting with typical symptoms of COVID-19.

… it is unclear from this narrative how the authors have reconciled these polarised recommendations into the offered, “low recommendation”. How were the final recommendations reached on the basis of conflicting statements?  

Further, it appears that the strength of the recommendations, the so called SRC scores, were derived on the basis of a response rate of just 23% from participating societies? If so, I question the validity of these classifications. Was any other methodology used to assist in defining the strength of the recommendations?

It seems unbelievable to me that there is any disagreement to the suggestion (recommendation 10) that “All patients should be screened for SARS-CoV-2 by nasopharyngeal swabs”, yet this recommendation is only graded “low”; (again) it is unclear why this is so?

I encourage the authors to publish more details of both methodology and results (as supplementary content if necessary) to allow the reader to better understand how conclusions were reached and to provide confidence in the robustness of the outcome.

In many cases I find the recommendations to be to excessively black and white. “19 Avoid high-flow nasal oxygen or aerosol-generating procedures...” is a good example. Life is not so black and white! Surely on the basis of consideration of an individual patient’s potential to benefit vs risk to staff with appropriate risk mitigation in place it may be reasonable on occasion to use high flow nasal oxygen, or to perform a diagnostic AGP – e.g. bronchoscopy?

Much of the discussion simply repeats many of the consensus recommendations without adequately exploring points of interest, relevant controversies, or knowledge gaps. This is a major shortage requiring a significant re-write.

The limitations of the methodology used (and indeed the low response rate) should be considered in the discussion section.

Minor comments

The term PPE2 requires explanation.

  1. The use of laminar air flow is recommended. Please clarify. Many authors are suggesting negative pressure theatres where available to protect staff in other areas, rather than extreme dispersion of theatre air in the surrounding environment.

19. The statement “various filters and attachments have been recommended” is wholly inadequate without explanation or referencing.

Author Response

Dear Editorial Board and Reviewers,

Thank you very much for the valuable review and the possibility to provide a revised version of our manuscript "Thoracic surgery in the COVID-19 pandemic: A novel approach to reach guideline consensus" (Manuscript ID: JCM-1176808-R1) to be considered for publication in your esteemed Journal of Clinical Medicine as an original article.

First, we would like to thank the reviewers for the valuable comments, which have been very fruitful for the refinement of our manuscript. The entire manuscript was thoroughly revised according to the referees’ comments. A detailed point-by-point response is attached below. We assume that this revision has substantially improved the quality of the manuscript and hope that the reviewers and editorial board will agree.

All authors are in agreement with the content of the revised version. The content has not been published or is under consideration for publication elsewhere.

Thanking you in advance for your endeavors.

Yours sincerely,

Tomasz Dziodzio, MD

Point-by-point response to the comments:

Response to the summary of assessments by Reviewers:

Reviewer 2:

Thank you for the opportunity to review. I read with interest this manuscript attempting to distil some consensus from a broad range of specialist society guidelines concerning the practice of Thoracic surgery during the COVID-19 pandemic. The sheer quantity and breadth of guidelines and recommendations that have been rapidly generated in response to the pandemic has been vast, and I am therefore convinced that there is a real place for the type of summary publication presented, particularly where the methodology is robust.

Major comments

The manuscript brings together the recommendations of many international thoracic surgical societies, but regrettably has failed to acknowledge the existence of a similar number of anaesthetic / perioperative publications, the subject mater of which overlap significantly.

Response: We thank the reviewer for this deep analysis of our manuscript and the remark. We absolutely agree that there are many recommendations on this topic from other sources, especially from anesthesiological societies, which clearly overlap with our manuscript. However, in order not to lose focus, we concentrated primarily on the thoracic surgical societies. Therefore, this manuscript does not claim to be complete and is primarily addressed to surgeons as readership. We acknowledged this limitation in the discussion. Furthermore, we have partially rewritten the discussion and analyzed corresponding recommendations of other medical societies and added them to the discussion.

I strongly support the attempt to use robust methodology to distil consensus from a number of similar, but at times divergent sources. The methodology however is inadequately described. Further, the results of this process are inadequately described, and the reader is left uncertain in places as to how any particular recommendation has been reached. The following is a prime (but not the only) example:

3.1.2 Staff

Staff members should not be screened for SARS-CoV-2.(SRC C) Two societies recommended to screen all hospital staff for SARS-CoV-2 and four so[1]cities disagreed on the topic. Some societies recommended nasopharyngeal swab tests in staff presenting with typical symptoms of COVID-19.

… it is unclear from this narrative how the authors have reconciled these polarised recommendations into the offered, “low recommendation”. How were the final recommendations reached on the basis of conflicting statements? 

Further, it appears that the strength of the recommendations, the so called SRC scores, were derived on the basis of a response rate of just 23% from participating societies? If so, I question the validity of these classifications. Was any other methodology used to assist in defining the strength of the recommendations?

Response: We thank the reviewer for this comment. The aim of this manuscript was not to provide clear recommendations, but to summarize available core topics from a multitude of existing statements (with different actuality and diverging recommendations for action). This core topics can later serve as a basis for generating societal recommendations. The process of recommendation generation can be very time-consuming and complex especially in a constantly changing pandemic situation (numbers of cases, availability of testing facilities and constant changes in procedural instructions) and recommendations can be outdated at time of publication. Elaborate classical guideline development strategies such as the Delphi or CORE method are not applicable at the current situation. Therefore, we used a modified Delphi template for our SRC building and set a high threshold for our recommendations. Key statements and responses of the societies were summarized and we calculated the SRC by subtracting the number of negative recommendations from the count of positive recommendations. Zero points were counted if the answer was left open or no statement was available [“strong recommendation” (SRC=A, score >10, societal approval of >70%), “medium recommendation” (SRC=B, score 4 to 9, societal approval of 25-69%) or “low recommendation” (SRC=C, score 1 to 3, societal approval of <25%)]. 6 out 14 (43%) societies commented on the topic of staff screening. Two agreed to screen all staff and 4 disagreed. In accordance with our score this yields a rate of 23% and thus the recommendation is to be considered low, as there is a clear disagreement here. Hence, we consider the score to be adequate for this analysis; of course, we agree with the reviewer and are aware that a much more robust score is necessary for a further and more detailed elaboration of consensus building. Additionally, it is worth mentioning that many societies have not updated their recommendations for some time and, that staff and inpatient screening has become far easier now than it was three months ago. Therefore, it can be assumed that this statement would be different today. We have now addressed this aspect in the discussion (page 11, line 363).

It seems unbelievable to me that there is any disagreement to the suggestion (recommendation 10) that “All patients should be screened for SARS-CoV-2 by nasopharyngeal swabs”, yet this recommendation is only graded “low”; (again) it is unclear why this is so?

Response: We thank the reviewer for this important comment. The comment addresses the screening recommendation of inpatients. The situation is the same as abovementioned staff screening. The timeframe for this manuscript was March 2020 and went through March 31st, 2021. During this timeframe some recommendations were modified and changed. However, many societies have failed to adapt their recommendations to the current situation. Therefore, the recommendation is only graded “low” although screening of staff and inpatients has become standard procedure. We addressed this aspect in the discussion (page 11, line 363 and page 12 line 384) 

I encourage the authors to publish more details of both methodology and results (as supplementary content if necessary) to allow the reader to better understand how conclusions were reached and to provide confidence in the robustness of the outcome.

In many cases I find the recommendations to be to excessively black and white. “19 Avoid high-flow nasal oxygen or aerosol-generating procedures...” is a good example. Life is not so black and white! Surely on the basis of consideration of an individual patient’s potential to benefit vs risk to staff with appropriate risk mitigation in place it may be reasonable on occasion to use high flow nasal oxygen, or to perform a diagnostic AGP – e.g. bronchoscopy?

Much of the discussion simply repeats many of the consensus recommendations without adequately exploring points of interest, relevant controversies, or knowledge gaps. This is a major shortage requiring a significant re-write.

The limitations of the methodology used (and indeed the low response rate) should be considered in the discussion section.

 Response: We thank the reviewer for these remarks. We absolutely agree that decision-making in the pandemic is no straight-forward and there is rarely one right way. Each country, each region, each hospital has a different pandemic situation and different resources to deal with the pandemic. Therefore, some recommendations may apply well for some readers other maybe not. We agree with the reviewer that there can´t be a “one fits all” solution. Additionally, the development of the pandemic also plays a relevant role as the actuality of each recommendation may vary over time. Therefore, we have tried to adopt most of the existing recommendations from thoracic surgeon societies as strictly as possible, which sometimes results in a black and white painting of our manuscript. Due to the reviewer's valuable comments, we have now addressed this fact more precisely in our discussion. Additionally, we absolutely understand the reviewer's assessment that there is a lack of high-quality statements in some parts of this manuscript, but it is also difficult to extract those statements from the large number of existing recommendations/publications in the focused field of thoracic surgery that reflect the actual situation of the pandemic and at the same time show a high level of scientific evidence. However, we hope that our revision now meets the reviewer's expectations.

Minor comments

The term PPE2 requires explanation.

Response: We thank the reviewer for this comment. The term “PPE” with its three categories was now clearly defined in the manuscript.

The use of laminar air flow is recommended. Please clarify. Many authors are suggesting negative pressure theatres where available to protect staff in other areas, rather than extreme dispersion of theatre air in the surrounding environment.

Response: We thank the reviewer for this comment. Several techniques for theatre ventilation already exist (laminar air-flow, negative pressure environments), however, current evidence on the efficacy of certain techniques to reduce the risk of transmission of SARS-CoV-2 in operating or procedural rooms is lacking [1]. To this date, literature only suggests the prevention of airborne contamination, namely in orthopedic surgery [2]. Still, several authors recommend negative pressure environments as a preventive measure for protection against SARS-CoV-2 [3-6]. We edited statement 16 in order to incorporate these thoughts.

  1. The statement “various filters and attachments have been recommended” is wholly inadequate without explanation or referencing.

Response: We thank the reviewer for this comment. We agree with the reviewer and substantiated our statement. As of today, the consensus acquired by our study on the basis of thoracic surgery societies remains the same and high-flow nasal oxygen (HFNO) is still not recommended due to fear of aerosol dispersion. However, recent evidence on aerosol generation associated with HFNO fails to establish a clear link between the use of HFNO and infection of healthcare workers with SARS-CoV-2 [7,8]. On the other hand, HFNO has been proven to reduce intubation rates and overall mortality in patients with COVID-19 [9,10]. Therefore, proven benefits and potentially unknown risks associated with HFNO must be carefully balanced. A simple tool to minimize aerosol generation may be a surgical mask [11].

References:

  1. Theodorou, C.; Simpson, G.S.; Walsh, C.J. Theatre ventilation. Ann R Coll Surg Engl 2021, 103, 151-154, doi:10.1308/rcsann.2020.7146.
  2. Thomas, A.M.; Simmons, M.J. The effectiveness of ultra-clean air operating theatres in the prevention of deep infection in joint arthroplasty surgery. Bone Joint J 2018, 100-B, 1264-1269, doi:10.1302/0301-620X.100B10.BJJ-2018-0400.R1.
  3. Flemming, S.; Hankir, M.; Ernestus, R.I.; Seyfried, F.; Germer, C.T.; Meybohm, P.; Wurmb, T.; Vogel, U.; Wiegering, A. Surgery in times of COVID-19-recommendations for hospital and patient management. Langenbecks Arch Surg 2020, 405, 359-364, doi:10.1007/s00423-020-01888-x.
  4. Sadrizadeh, S.; Holmberg, S. Surgical clothing systems in laminar airflow operating room: a numerical assessment. J Infect Public Health 2014, 7, 508-516, doi:10.1016/j.jiph.2014.07.011.
  5. Al-Benna, S. Negative pressure rooms and COVID-19. J Perioper Pract 2021, 31, 18-23, doi:10.1177/1750458920949453.
  6. Gonzalez-Ciccarelli, L.F.; Nilson, J.; Oreadi, D.; Fakitsas, D.; Sekhar, P.; Quraishi, S.A. Reducing transmission of COVID-19 using a continuous negative pressure operative field barrier during oral maxillofacial surgery. Oral Maxillofac Surg Cases 2020, 6, 100160, doi:10.1016/j.omsc.2020.100160.
  7. Agarwal, A.; Basmaji, J.; Muttalib, F.; Granton, D.; Chaudhuri, D.; Chetan, D.; Hu, M.; Fernando, S.M.; Honarmand, K.; Bakaa, L.; et al. High-flow nasal cannula for acute hypoxemic respiratory failure in patients with COVID-19: systematic reviews of effectiveness and its risks of aerosolization, dispersion, and infection transmission. Can J Anaesth 2020, 67, 1217-1248, doi:10.1007/s12630-020-01740-2.
  8. Li, J.; Fink, J.B.; Ehrmann, S. High-flow nasal cannula for COVID-19 patients: low risk of bio-aerosol dispersion. Eur Respir J 2020, 55, doi:10.1183/13993003.00892-2020.
  9. Demoule, A.; Vieillard Baron, A.; Darmon, M.; Beurton, A.; Geri, G.; Voiriot, G.; Dupont, T.; Zafrani, L.; Girodias, L.; Labbe, V.; et al. High-Flow Nasal Cannula in Critically III Patients with Severe COVID-19. Am J Respir Crit Care Med 2020, 202, 1039-1042, doi:10.1164/rccm.202005-2007LE.
  10. Patel, M.; Gangemi, A.; Marron, R.; Chowdhury, J.; Yousef, I.; Zheng, M.; Mills, N.; Tragesser, L.; Giurintano, J.; Gupta, R.; et al. Retrospective analysis of high flow nasal therapy in COVID-19-related moderate-to-severe hypoxaemic respiratory failure. BMJ Open Respir Res 2020, 7, doi:10.1136/bmjresp-2020-000650.
  11. Leonard, S.; Atwood, C.W., Jr.; Walsh, B.K.; DeBellis, R.J.; Dungan, G.C.; Strasser, W.; Whittle, J.S. Preliminary Findings on Control of Dispersion of Aerosols and Droplets During High-Velocity Nasal Insufflation Therapy Using a Simple Surgical Mask: Implications for the High-Flow Nasal Cannula. Chest 2020, 158, 1046-1049, doi:10.1016/j.chest.2020.03.043.

Reviewer 3 Report

The paper deal with an current problem of great clinical relevance. With the availiable recommendations of experts the authors provide a summary that can help with decision making during the COVID pandemic.

Publication is to recommend.

Author Response

Dear Editorial Board and Reviewers,

Thank you very much for the valuable review and the possibility to provide a revised version of our manuscript "Thoracic surgery in the COVID-19 pandemic: A novel approach to reach guideline consensus" (Manuscript ID: JCM-1176808-R1) to be considered for publication in your esteemed Journal of Clinical Medicine as an original article.

First, we would like to thank the reviewers for the valuable comments, which have been very fruitful for the refinement of our manuscript. The entire manuscript was thoroughly revised according to the referees’ comments. A detailed point-by-point response is attached below. We assume that this revision has substantially improved the quality of the manuscript and hope that the reviewers and editorial board will agree.

All authors are in agreement with the content of the revised version. The content has not been published or is under consideration for publication elsewhere.

Thanking you in advance for your endeavors.

Yours sincerely,

Tomasz Dziodzio, MD

Reviewer 3:

The paper deal with a current problem of great clinical relevance. With the availiable recommendations of experts the authors provide a summary that can help with decision making during the COVID pandemic.

Response: We thank the reviewer for this assessment.

Round 2

Reviewer 1 Report

I find all the modifications and additions made appropriate

Author Response

We thank the reviewer for this assessment.

Reviewer 2 Report

Thank you for the opportunity to re-review. The authors have substantially edited their initial submission and as a result the manuscript is better balance and clearer. I would like to see the obvious limitation of the low response rate (23%) more explicitly highlighted in the discussion but otherwise have no further comments.

Author Response

We thank the reviewer for this feedback. Indeed, the response rate after contacting thoracic surgery societies and asking for their opinion of our proposed statements could have been better. By directly evaluating the statements, more clear deductions could have been possible. The discussion section was edited to incorporate this limitation.